# Downregulating Mitochondrial DNA Polymerase γ in the Muscle Stimulated Autophagy, Apoptosis, and Muscle Aging-Related Phenotypes in *Drosophila* Adults

**DOI:** 10.3390/biom12081105

**Published:** 2022-08-11

**Authors:** Mika Ozaki, Tuan Dat Le, Yoshihiro H. Inoue

**Affiliations:** Biomedical Research Center, Kyoto Institute of Technology, Matsugasaki, Sakyo-ku, Kyoto 606-8585, Japan

**Keywords:** mitochondria, DNA polymeraseγ, *Drosophila*, muscle aging, autophagy

## Abstract

Reactive oxygen species, generated as by-products of mitochondrial electron transport, can induce damage to mitochondrial DNA (mtDNA) and proteins. Here, we investigated whether the moderate accumulation of mtDNA damage in adult muscles resulted in accelerated aging-related phenotypes in *Drosophila*. DNA polymerase γ (Polγ) is the sole mitochondrial DNA polymerase. The muscle-specific silencing of the genes encoding the polymerase subunits resulted in the partial accumulation of mtDNA with oxidative damage and a reduction in the mtDNA copy number. This subsequently resulted in the production of abnormal mitochondria with reduced membrane potential and, consequently, a partially reduced ATP quantity in the adult muscle. Immunostaining indicated a moderate increase in autophagy and mitophagy in adults with RNA interference of *Polγ* (*PolγRNAi*) muscle cells with abnormal mitochondria. In adult muscles showing continuous silencing of Polγ, malformation of both myofibrils and mitochondria was frequently observed. This was associated with the partially enhanced activation of pro-apoptotic caspases in the muscle. Adults with muscle-specific PolγRNAi exhibited a shortened lifespan, accelerated age-dependent impairment of locomotor activity, and disturbed circadian rhythms. Our findings in this *Drosophila* model contribute to understanding how the accumulation of mtDNA damage results in impaired mitochondrial activity and how this contributes to muscle aging.

## 1. Introduction

Aging is characterized by multiple factors, including epigenetic changes, loss of proteostasis, cellular senescence, stem cell depletion or hyperproliferation, genomic instability, and mitochondrial dysfunction [1]. The progression of aging is closely related to oxidative stress and the presence of free radicals—molecular species with unpaired electrons. The free radical hypothesis was proposed as one major cause of aging progression five decades ago [2]. Recently, researchers have focused on reactive oxygen species (ROS), which are highly reactive molecules that contain free radicals. The ROS level is considered critical in determining lifespan [3]. As approximately 90% of intracellular ROS are produced in the mitochondria [4], how mitochondrial DNA (mtDNA) and proteins are damaged and how this contributes to aging remain topics of interest. Muscle consumes much more ATP than other tissues, and this ATP is supplied by the mitochondria. Thus, muscle cells possess many more mitochondria than cells in other tissues. In aged animals, muscle cells contain more mitochondria with reduced activity owing to accumulated oxidized mtDNA [5]. Therefore, considering the hyperaccumulation of DNA mutations in mitochondria, which are the main ROS production sites, the mitochondrial theory of aging has been proposed [2,6,7]. This theory suggests that ROS production in mitochondria increases with age, arising from the age-dependent decline in mitochondrial respiratory activity and antioxidative enzymes that remove ROS [8]. 

Unlike genomic DNA in the nucleus, a mitochondrion contains multiple copies of circular double-stranded DNA encoding genes for the major components of respiratory chain complexes [9]. As mtDNA is more directly exposed to ROS generated in organelles, it is more susceptible to damage from ROS than nuclear DNA [10,11]. Damage to mtDNA impairs its translational capacity, resulting in an insufficient supply of mtDNA-encoded proteins. This leads to the reduced efficiency of the respiratory chain, facilitating additional ROS production, which leads to mitochondrial dysfunction [8,12]. In other words, the increased levels of ROS generated during ATP synthesis result in a vicious cycle that causes further damage to mtDNA [13]. The accumulation of mtDNA mutations in somatic tissues accelerates with aging [14]. 

*Drosophila* Polγ is a heterodimer composed of the accessory subunit PolG2 and catalytic subunit PolG1 [15]. This polymerase is the only DNA Pol responsible for mtDNA replication and repair. A second polymerase, PrimPol, required for mtDNA damage, is not conserved in *Drosophila* [16]. Base excision repair (BER), a DNA repair process, is involved in the removal of oxidized bases, such as 8-oxo-deoxyguanine (8-oxodG), from mtDNA. However, the involvement of mammalian Polβ in mitochondrial BER has not been verified in *Drosophila* [17]. A mouse strain expressing a mutant mtDNA Polγ with compromised proofreading exonuclease function causes mtDNA mutations to quickly accumulate with aging [18]. Some studies have reported that these mice exhibit numerous phenotypes mimicking aging [18,19]. In contrast, aging-related phenotypes, including many disorders and traits that are otherwise not observed during normal aging, have also been noticed [20]. Other studies have shown that PolG-mutant mice harbor somatic point mutations at much higher frequencies. Even in *Drosophila*, controversial results regarding mtDNA mutations and age-related phenotypes have been reported [21,22]. 

Once damage occurs to biomolecules such as DNA and proteins in the mitochondria, the damaged part of the organelle is separated by fission and removed by mitophagy [23]. This regulation is essential for maintaining organelle homeostasis [24]. Subsequently, following the loss of mitochondria, new mitochondria are produced from existing mitochondria. To compensate for damaged mtDNA and oxidized proteins, mtDNA replication and *de novo* protein synthesis in organelles are stimulated [25]. After the fission of damaged mitochondria, the protein kinase PINK1 accumulates on the outer membrane of organelles. Subsequently, PINK1 recruits a ubiquitin ligase, parkin, to the outer membrane and, subsequently, activates parkin by phosphorylation [26]. Activated parkin associates ubiquitin with the outer membranes of damaged mitochondria. Adapter protein Ref(2)P binds to poly-ubiquitin chains marked by parkin on the surface of mitochondria and interacts with Atg8, which is composed of autophagosomes; thereafter, damaged mitochondria are taken up by autophagosomes [27,28]. The autophagosomes fuses with lysosomes to decompose damaged mitochondria [29]. PINK1-parkin-dependent mitophagy is essential for mitochondrial quality control [30]. In *Drosophila*, enhanced mitochondrial fission stimulates mitophagy, contributing to the maintenance of mitochondrial function in indirect flight muscles (IFMs) of aged adults [31]. However, how the interaction between mitophagy and mitochondrial dynamics influences age-related phenomena in muscles remains unclear.

In this study, we investigated whether the RNA-mediated silencing of two Polγ subunit genes in *Drosophila* adult muscles influences the aging-dependent phenotype. After we confirmed the efficient silencing of *Polγ* subunit genes, we demonstrated that oxidative DNA damage repair and mtDNA replication processes were moderately inhibited in the mitochondria of the adult muscle harboring RNA interference of *Pol**γ* (*Pol**γRNAi*). Consequently, the downregulation of DNA Polγ resulted in mitochondrial dysfunction. Furthermore, we obtained evidence indicating that mitochondrial fission was stimulated in *Polγ*-depleted muscles. Next, we investigated whether autophagy and apoptosis were also stimulated in *Pol**γRNAi* muscle. Myofibril and mitochondrial malformations are thought to be consequences of apoptosis, and we examined whether these occurred in adult muscles along with the continuous downregulation of DNA Polγ using transmission electron microscopy (TEM). From these results, we concluded that the muscle-specific reduction in *Polγ* expression in *Drosophila* adults moderately inhibited mtDNA replication and repair and reduced mitochondrial activity. Mitochondrial fission and apoptosis were enhanced in muscle cells. The lifespan of adults harboring muscle-specific *PolγRNAi* was shortened. Furthermore, the age-dependent decline in locomotor activity accelerated with aging in RNAi adults, and a perturbation of the circadian rhythm was also observed in aged adults. Based on the genetic evidence obtained from this study using the *Drosophila* model of accumulated mtDNA damage in IFM cells, we discuss whether the accumulation of mtDNA damage results in the impairment of mitochondrial activity and how this leads to muscle aging. 

## 2. Materials and Methods

### 2.1. Fly Stocks and Culture

We used Canton S as normal control stock. To induce the ectopic expression of target genes in specific tissues, the GAL4/UAS system was used [32]. To restrict the Gal4 activation in the adult stage, a temperature-sensitive mutant of the *Gal80* gene, which encodes an inhibitor of the Gal4 protein, was used. The mutant proteins were specifically inactivated in the adult stage by transferring the flies at a non-permissive temperature, 28 °C. *P{tubP-GAL80^ts^}*; *P{GAL4-Mef2.R}R1* (#67063) (hereinafter referred to as *Mef2^ts^-Gal4*) was used for the ectopic expression of muscle cell-specific genes located downstream of the UAS sequences [33]. The following *UAS-RNAi* lines were used for RNAi-based gene silencing experiments: *P{GD17669}v49765* (#v49765) (*UAS-PolG2RNAi^1^*), *P{y^+t7.7^ v^+t1.8^* = *TRiP.HMS05747}attP40* (#67925) (*UAS-PolG2RNAi^2^*), *P{y^+t7.7^ v^+t1.8^* = *TRiP.JF01532}attP2* (#31081) (*UAS-PolG1RNAi^1^*), and *P{y^+t7.7^ v^+t1.8^* = *TRiP.JF01563}attP2* (#31098) (*UAS-PolG1RNAi^2^*). These *UAS-RNAi* stocks, except *UAS-PolG2RNAi^1^* (from Vienna *Drosophila* Resource Center [Vienna, Austria]), were obtained from the Bloomington *Drosophila* Stock Center (Bloomington, IN, USA). 

To maintain the stocks and obtain adults for aging-related experiments, the following standard cornmeal fly food was prepared: 40 g of dried yeast (Asahi Breweries, Ltd. Tokyo, Japan), 40 g of cornmeal, 100 g of glucose, and 7.2 g of powdered agar (agarose) were added to 1 L of distilled water and heated with stirring. After cooling below 75 °C, 5 mL of 10% methyl parahydroxybenzoate in 70% ethanol and 5 mL of propionic acid (Fujifilm Wako Pure Chemical Co., Osaka, Japan) were added as preservatives. The fly diet mixture was dispensed into a plastic vial (diameter 22 mm, height 96 mm) (Chiyoda Science Co., Tokyo, Japan) and filled with a sponge plug for use. Fly stocks were maintained and genetic crosses were performed at 25 °C. For silencing of the target genes in the adult stage using the *Gal4^ts^* lines, individuals were reared at 19 °C until eclosion. Newly eclosed adults were collected and transferred to the incubator at 28 °C. 

For drug administration, the instant fly food prepared from 1 mL of water and 0.3 g of Instant Medium (Formulas (4)–(24), Blue) (Carolina Biological Supply Company, Burlington, IN, USA) was used. To cause oxidative stress, 10 mM paraquat (PQ) (1,1′-dimethyl-4,4′-dipyridinium chloride) (Fujifilm Wako Pure Chemical Co. Osaka, Japan) was added to the instant fly food. 

### 2.2. Immunofluorescence

IFMs were prepared from the adult thoraxes dissected in relaxing buffer (0.1 M KCl, 20 mM Tris-HCl, pH = 7.2, 1 mM MgCl_2_, 1 mM EDTA) [34]. The specimens were fixed in 4% paraformaldehyde for 30 min. After washing with 0.1% PBST (1× PBS, 0.1% TritonX-100) and subsequent blocking in 10% normal goat serum, the primary antibody diluted with the blocking solution was added and incubated at 4 °C overnight. The following primary antibodies were used: anti-Atg8 antibody (#ab109364, Abcam, Cambridge, UK) (dilution 1/400), anti-Ref(2)P antibody (#ab178440, Abcam) (1/500), anti-ATP5A antibody (#ab14748, Abcam) (1/400), and anti-Cleaved Drosophila Dcp-1 antibody (#9578, Cell Signaling Technology, Inc., Danvers, MA, USA) (1/100). 

After washing IFM samples with 0.1% PBST, Alexa fluorescence dye-conjugated secondary antibodies (Invitrogen, Waltham, MA, USA) were incubated with the samples for 2 h. For the visualization of F-actin, Alexa Fluor 488-conjugated phalloidin (#A12379, Invitrogen) was added simultaneously. After washing with 0.1% PBST several times, the specimens were embedded with VECTASHIELD Mounting Medium (Vector Laboratories, Burlingame, CA, USA) and observed using a laser scanning confocal microscope (FV10i, Olympus, Tokyo, Japan). Images were acquired at 512 × 512 pixel size. For image processing, FV10-ASW 4.2 Viewer (Olympus) was used.

### 2.3. Quantification of 8-oxo-dG in Mitochondrial DNA

IFMs were fixed for 5 min with Carnoy’s solution (100% ethanol: acetic acid = 3:1) [35]. Fixed IFMs were incubated for 10 min in 2 N HCl to denature dsDNA. After repeated washing, neutralization in Tris-HCl (pH 8.0), and blocking with 10% normal goat serum, fixed specimens were incubated overnight at 4 °C with an anti-8-oxo-dG antibody (#4354-MC-050, Trevigen Inc., Gaithersburg, MD, USA) (1/800). After repeated washing with 0.1% PBST, the samples were incubated with Cy3-conjugated anti-mouse secondary antibody (#115-165-062, Jackson ImmunoResearch Inc., West Grove, PA, USA) (1/400). To visualize mitochondrial DNA, IFMs were treated with Quant-iT PicoGreen (#P7581, Invitrogen) (1/200) for 2 h. The samples were mounted and observed as described above. To quantify mitochondrial 8-oxo-dG foci, ImageJ (NIH, Bethesda, MD, USA) was used. To avoid measuring 8-oxo-dG signals from nuclear DNA, we selected immunofluorescence-positive foci with a size of 10 pixels or fewer for quantification. 

ImageJ ver. 1.52a was used to quantify the distribution of relevant proteins visualized by immunostaining. To quantify anti-Ref(2)P immunofluorescence foci in adult muscles, the total fluorescence intensity of foci larger than 10 pixels was selected in each fluorescent image (512 × 512 pixels) (IFM cross-section 4.0 × 10^−2^ mm^2^). Ten pixels of the fluorescent aggregates were calculated as one unit, and the fractions below the decimal point were discarded. The length of each mitochondrion was measured using ImageJ, and the average value per confocal microscopic field (IFM cross-section 4.4 × 10^−2^ mm^2^) was calculated. Anti-cDcp1 and anti-Atg8 immunostaining foci were quantified by measuring the total area exhibiting the immunostaining signals per confocal microscopic field (4.4 × 10^−2^ mm^2^ of the IFM cross-section). Finally, mitophagy in adult muscle was quantified by selecting the anti-Atg8 immunostaining foci that overlapped with anti-mitochondrial ATPase V immunostaining signals in the IFM optical field.

### 2.4. Tetramethylrhodamine Ethyl Ester (TMRE) Staining

For the quantification of mitochondrial membrane potential in adult muscle, IFMs were collected from adult thoraxes and dispensed in the relaxing buffer. They were incubated with 100 nM TMRE (#T669, Invitrogen) for 20 min and fixed in 4% paraformaldehyde for 40 min. The specimens were observed within 1 h after fixation using a laser scanning confocal microscope (Fv10i, Olympus), and images of 512 × 512 pixels were acquired. To quantify TMRE fluorescent intensity, ImageJ was used. Three points on each mitochondrion were selected to calculate the average fluorescence intensity per confocal microscopic field.

### 2.5. Quantitative Reverse Transcription-Polymerase Chain Reaction (RT-PCR)

Total RNA was prepared from thoraxes and legs of 30 flies using TRIzol reagent (#15596026, Invitrogen). After the extraction of the homogenates with CIAA (chloroform: isoamyl alcohol = 24:1), nucleic acid contained in the aqueous phase was precipitated by isopropanol. Traces of DNA were removed by DNase I (Rnase-Free DNase I, #D9905K, Epicentre/Lucigen, Middleton, WI, USA) treatment for 30 min at 37 °C. Subsequently, phenol/CIAA (phenol:chloroform:isoamyl alcohol = 25:24:1) was added to inactivate DNase. The RNA contained in the solution was precipitated by adding isopropanol and collected by centrifugation. Using RNA as a template, cDNA was synthesized using a PrimeScript II 1st strand cDNA Synthesis Kit (#6210A, Takara Bio Inc., Shiga, Japan) with a random primer. RNA extraction and cDNA synthesis were independently repeated three times per genotype.

qRT-PCR was performed using FastStart Essential DNA Green Master (#06402712001, Roche Diagnostics, Mannheim, Germany). qPCR was performed as three independent reactions using LightCycler Nano (#06407773001, Roche Diagnostics). After the denaturation of the template DNA at 95 °C for 10 min; 45 cycles of 95 °C for 10 s, 60 °C for 10 s, and 72 °C for 15 s were repeated. After the final elongation reaction at 72 °C for 30 s, melting curve analysis was performed at temperatures of 60 to 95 °C at 0.1 °C/s. Quantitative analysis was performed using the ΔΔCq method and *RP49* was used as a reference for normalization [36]. The following primers were used:
RP49F: TTCCTGGTGCACAACGTG
R: TCTCCTTGCGCTTCTTGGPolG2F: CTTCTACAACATGCAGCGTGAG
R: TAGCTCGTGCGGATATCGATGPolG1F: ACAATGTCGCTGCACATGTG
R: CCTTCTTGGATTTGAGCATGGCCOX IIIF: TGACCATTAACAGGAGCTATCGG
R: CCTTCTCGTGATACATCTCGTCA

### 2.6. Nucleic Acid Preparation for Quantitative PCR to Estimate mtDNA Copy Number

A nucleic acid solution containing genomic DNA was prepared from adult muscle. Thoraxes and legs collected from adults in the relaxing buffer were homogenized in DNA extract solution (100 mM Tris-HCl, pH = 8.0, 0.5% SDS, 50 mM NaCl, 100 mM EDTA). The tissue homogenates were treated with Proteinase K (#164-14004, Fujifilm Wako Pure Chemical Co.) at 55 °C for 1 h. After the extraction of the homogenates with Phenol/CIAA (phenol:chloroform:isoamyl alcohol = 25:24:1) to denature proteins, the aqueous phase containing nucleic acids was collected. The nucleic acids contained in the phase were precipitated by adding ethanol. The precipitates containing genomic DNA were used for quantitative PCR to estimate the ratio of mitochondrial DNA and nuclear DNA.

### 2.7. ATP Assay

Five adult thoraxes were homogenized in 1% PBST on ice. These extracts were immediately frozen in liquid nitrogen and, subsequently, inactivated at 99 °C for 3 min. After centrifugation at 6000× *g* for 10 min, the ATP levels were quantified in the supernatants using the ATP Determination Kit (#A22066, Invitrogen). The fluorescence intensity was measured using a luminometer (Lumat LB9507, Berthold Technologies, Bad Wildbad, Germany). Based on the standard curve created, the ATP levels of the samples were determined. Protein concentration was measured using the SmartSpec Plus Spectrophotometer (#1702525JEDU, Bio-Rad, Hercules, CA, USA).

### 2.8. Transmission Electron Microscope Observation of IFMs

Thoraxes collected from 36-d-old flies were placed in fixative solution (4% paraformaldehyde, 2% glutaraldehyde in 0.1 M cacodylate buffer, pH = 7.4). These thoraxes were prepared for TEM, as described previously [37]. Briefly, the thoraxes were washed with 0.1 M cacodylate buffer and fixed with 2% osmium tetroxide in 0.1 M cacodylate buffer. The specimens were dehydrated by consecutive incubation in 50% ethanol, 70% ethanol, 90% ethanol, and 100% ethanol. After the specimens were infiltrated with propylene oxide and put into a 7:3 mixture of propylene oxide and resin (Quetol-812, Nisshin EM Co., Tokyo, Japan), they were transferred to new 100% resin and polymerized. The polymerized resins were ultrathin sectioned at 70 nm using Ultracut-UCT (Leica, Vienna, Austria) and mounted on copper grids. After staining with 2% uranyl acetate, the sections were washed with distilled water and stained with lead stain solution (Sigma-Aldrich Co., Tokyo, Japan). The grids were observed using a transmission electron microscope (JEM-1400Plus, JEOL Ltd., Tokyo, Japan) at 100 kV acceleration voltage and photographed with a CCD camera (EM-14830RUBY2, JEOL Ltd.).

### 2.9. Survival Assay

Adult males were collected within 1 day after eclosion under CO_2_ anesthesia. Next, 10 to 20 flies in each trial (*n* > 103 flies in total) were reared at 28 °C in a plastic vial containing instant fly food (see above). The number of dead flies was counted every 24 h. Adults were subsequently transferred to new vials containing fresh food every 4 d. The survival rate was calculated by the Kaplan-Meier method using GraphPad Prism 6 (GraphPad software Co., San Diego, CA, USA).

### 2.10. Climbing Assay

The locomotor activity of adults was quantified using the climbing assay [38] as described previously [36]. Ten to twenty adult males were collected in empty plastic vials (22 mm in diameter, 96 mm in height; Chiyoda Science Co., Tokyo, Japan) and left for 10 min. Within 6 s of tapping the flies down to the bottom, the number of flies that climbed the vial wall was counted. The average score for each vial was calculated by counting the number of adults, with 10 points for adults above the 5 cm reference line, 5 points for adults between the bottom and the reference line, and 0 points for adults remaining at the bottom. Three trials were performed with an interval of at least 1 min, and the average value of the three trials was calculated. The assays were performed every 5 d from the first day after eclosion until the adult survival rate reached 50%.

### 2.11. Locomotor Assay

The daily locomotor activity of the adults was measured using a DAM2 *Drosophila* Activity Monitor (TriKinetics Inc., Waltham, MA, USA), following a previously published protocol [39]. A young male fly within 24 h of eclosion was reared in a measuring tube (diameter 7 mm, length 65 mm). The tubes were set on a monitor at 28 °C to ensure that infrared rays reached the center of the tube. Adults were transferred to new tubes containing fresh food every 4 d. The number of adults that crossed an infrared light beam for 30 min was counted using DAM System 308 software (TriKinetics Inc., Waltham, MA, USA). Locomotion was measured from the first day to 36 d after eclosion. The light in the incubator was on 09:00–21:00 and off 21:00–09:00. 

### 2.12. Statistical Analysis

The significance of differences in the survival curves between the groups was analyzed using the log-rank test. For the comparisons of the two groups, we used the Student’s *t*-test. One-way ANOVA followed by Bonferroni post-hoc test was applied to assess the differences in more than two groups. Two-way ANOVA followed by Tukey post-hoc was performed to compare the mean differences between groups that were split into two independent variables. Data were considered significant at *p*-values < 0.05. Statistical analyses were performed using GraphPad Prism (Version 9, GraphPad Software, San Diego, CA, USA).

## 3. Results

### 3.1. Adult Muscle-Specific Silencing of Polγ Subunit Genes Partially Inhibited Mitochondrial DNA Replication and Repair of Oxidatively Damaged DNA in Drosophila Adult Muscle

To simulate aging-related mtDNA damage accumulation in adult muscle, we silenced the mRNAs encoding the Polγ subunits PolG2 and PolG1 [40] in the muscle at the adult stage using double-stranded RNA (dsRNA)-mediated RNAi. First, we investigated how the mRNA levels of these two subunits changed as normal flies aged. We performed qRT-PCR to quantify the levels of *PolG2* and *PolG1* mRNAs in control adults (*Mef2^ts^* > *+*) 5, 20, and 36 d after eclosion. The level of *PolG2* mRNA in the thoraxes of control adults reared at 28 °C significantly declined with aging in 20- and 36-d-old adults (74% and 40% of the level in 5-d-old adults, respectively; *p* < 0.01 and *p* < 0.0001, respectively; one-way ANOVA with Bonferroni’s multiple comparisons test) (Figure 1A). In contrast, *PolG1* mRNA remained at a constant level, regardless of age (Figure 1D). 

We established an RNAi system that enabled the silencing of the DNA Polγ subunit genes at the mRNA level, specifically in adult muscles. We crossed the *Gal4^ts^* line carrying both *Mef2-Gal4* and *Gal80^ts^* with the *UAS-PolG2RNAi* or *UAS-PolG1RNAi* stocks. Total RNA was isolated from adult thoraxes to quantify the mRNA levels by qRT-PCR (*n* = 9). The dsRNAs against *PolG2* mRNA induced by *UAS*-*PolG2RNAi^1^* and *UAS*-*PolG2RNAi^2^* with *Mef2^ts^-Gal4* efficiently reduced mRNA levels to 66% and 34% of the control level in 5-d-old adults, respectively (Figure 1B). The levels further declined to 49% of the control level in 20-d-old adults of both *PolG2RNAi* types (Figure 1C). Similarly, the dsRNAs against *PolG1* mRNAs induced by *UAS*-*PolG1RNAi^1^* and *UAS*-*PolG1RNAi^2^* with *Mef2^ts^-Gal4* (*Mef2^ts^* > *UAS-PolG1RNAi^1^* and *Mef2^ts^* > *UAS-PolG1RNAi^2^*) reduced the mRNA levels in 20-d-old flies compared with 5-d-old flies (Figure 1E,F). We confirmed that the mRNA levels of DNA Polγ subunits were reduced in the RNAi adult muscles.

To investigate whether oxidative DNA damage to mtDNA increased owing to the downregulation of Polγ genes, we performed immunostaining of IFMs with an antibody that recognized 8-oxodG (*n* > 20) (Figure 2A). Adult males with the muscle-specific silencing of Polγ genes were fed 10 mM PQ to cause oxidative stress. Using anti-8-oxodG immunostaining, several immunostaining foci were observed in the cytoplasm of IFM cells harboring *PolG2RNAi or PolG1RNAi* (Figure 2Ab’–e’,g’–j’). The smaller 8-oxodG immunostaining foci likely corresponded to oxidized mtDNA. The total pixel size in the IFMs of flies fed PQ for 5 d after eclosion was comparable to that of control adults at the same age (*Mef2^ts^* > *+*) (*n* = 20). The total pixel size in the areas of flies fed PQ for 10 d increased, compared to that of control adults at the same age (*Mef2^ts^* > *+*) (*n* > 20). Although no significant differences were observed between every RNAi type and the control in both age groups (*p* > 0.05, one-way ANOVA with Bonferroni’s multiple comparisons test), there was a trend toward an increase in 8-oxodG foci (Figure 2B). These observations suggested that oxidative damage accumulated on the mtDNA in adults harboring *PolγRNAi* aged under external oxidative conditions (Figure 2B). 

A reduction in the mtDNA copy number is a characteristic feature of mitochondrial dysfunction that becomes apparent with age [41]. We performed qPCR to investigate whether mtDNA copy number in the thoraxes of adults harboring *PolγRNAi* decreased. In control adults (*Mef2^ts^* > *+*), there were no changes in the relative amount of mtDNA to nuclear DNA in the thoraxes between 5- and 36-d-old adults (*n* = 3, Student’s *t*-test) (Appendix A). However, in 36-d-old adults harboring PolγRNAi (*Mef2^ts^* > *PolG2RNAi^1^*), the amount of mtDNA significantly decreased, compared with that in control adults of the same age (*Mef2^ts^* > *+*) (*n* = 9) (*p* < 0.001, one-way ANOVA with Bonferroni’s multiple comparisons test), but not in other *PolγRNAi* types (*p* > 0.05) (Appendix A). These results suggest that both DNA repair and the replication of mtDNA were at least partially inhibited in the muscles of aged adults harboring *PolγRNAi*, especially *PolG2*. Therefore, these observations suggest that *PolγRNAi* results in the accumulation of abnormal mitochondria carrying damaged mtDNA in the adult muscle.

### 3.2. Reduced mRNA Levels of Polγ Genes Resulted in Partially Reduced Mitochondrial Activity in Adult Thoraxes Containing Flight Muscle as a Major Component

We examined whether the muscle-specific silencing of *Polγ* aggravated the age-related loss of mitochondrial activity in adult muscle. To estimate mitochondrial activity, we investigated mitochondrial membrane potential using TMRE as an indicator. We observed TMRE-stained IFMs using confocal microscopy and obtained typical images (*n* ≥ 20) (Figure 3A). The IFMs of 36-d-old control adults showed much lower fluorescence intensity than those of 5-d-old control adults (Figure 3Aa,f). These observations confirmed that the mitochondrial membrane potential visualized by TMRE was lost in the aged muscle. Thus, we investigated whether *PolγRNAi* changed mitochondrial membrane potential under the same conditions. In the IFM of 5-d-old adults harboring muscle-specific *Polγ*RNAi (*Mef2^ts^* > *PolG2RNAi^1^*, *Mef2^ts^* > *PolG2RNAi^2^*, and *Mef2^ts^* > *PolG1RNAi^1^*), the TMRE fluorescence intensity was significantly lower than that in the controls in every case (*Mef2^ts^* > *+*) (*p =* 0.0168, *p =* 0.0395, and *p =* 0.0134, respectively; one-way ANOVA with Bonferroni’s multiple comparisons test) (Figure 3B). All 36-d-old adults harboring *PolγRNAi* also exhibited slightly lower TMRE fluorescence intensity than 36-d-old control adults (*Mef2^ts^* > *+*), although there were no significant differences in the intensity between controls and *PolγRNAi* adults (*p* > 0.05, one-way ANOVA with Bonferroni’s multiple comparisons test) (Figure 3B). These observations suggested that *Polγ* silencing resulted in reduced mitochondrial membrane potential in the muscles of young adults.

The maintenance of mitochondrial membrane potential is critical for ATP production [42]. Thus, we investigated whether ATP production changed in adult thoraxes harboring *PolγRNAi*. We performed a luciferase assay to quantify the relative ATP levels in the adult thoraxes (*n* = 3). The average ATP levels in the muscle of 5- and 36-d-old adults harboring *PolγRNAi* were slightly lower than those in control adults at the same age (93% of the control in *Mef2^ts^* > *PolG2RNAi^1^*, 70% in *Mef2^ts^* > *PolG2RNAi^2^*, 102% in *Mef2^ts^* > *PolG1RNAi^1^*, and 94% in *Mef2^ts^* > *PolG1RNAi^2^*, respectively) (Appendix A) and (88% of the control in *Mef2^ts^* > *PolG2RNAi^1^*, 92% in *Mef2^ts^* > *PolG2RNAi^2^*, 97% in *Mef2^ts^* > *PolG1RNAi^1^*, and 86% in *Mef2^ts^* > *PolG1RNAi^2^*, respectively) (Appendix A). Although the differences were not significant between every type of *polγRNAi* adults and control adults (*p* > 0.05, one-way ANOVA with Bonferroni’s multiple comparisons test), these results were not inconsistent with the interpretation that muscle-specific *Polγ* silencing influenced mitochondrial activity in adult muscle.

### 3.3. Enhancement of Mitochondrial Fission in the IFMs Harboring PolγRNAi

Mitochondria containing damaged mtDNA, abnormal proteins, or oxidized lipids are efficiently removed by mitophagy to maintain homeostasis of the organelle [43]. Before mitophagy occurs, a damaged part of the abnormal mitochondria is separated by mitochondrial fission. Thus, we next investigated whether mitochondrial fission was enhanced in muscles harboring *PolγRNAi* containing dysfunctional mitochondria. We performed immunostaining of IFMs with an anti-ATP5A antibody to visualize the entire mitochondria (Figure 4A). The average length of mitochondria in the IFMs of 5-d old control flies was 2.83 μm (*n* = 20). However, the mitochondria in the 5-d-old adults harboring other types of *PolγRNAi* were significantly shorter than those in the controls of the same age (*p* < 0.0001, one-way ANOVA with Bonferroni’s multiple comparisons test) (Figure 4B). Consistently, in *PolG2RNAi* (*Mef2^ts^* > *PolG2RNAi^2^*) and *PolG1RNAi* (*Mef2^ts^* > *PolG1RNAi^1^* and *Mef2^ts^* > *PolG1RNAi^2^*) adults, the average length of mitochondria in the IFMs of 36-d-old adults harboring *PolγRNAi* was significantly shorter than that in control adults (*p* < 0.0001, one-way ANOVA with Bonferroni’s multiple comparisons test) (Figure 4B). The length of mitochondria, however, could not be measured in 36-d-old *Mef2^ts^* > *PolG2RNAi^1^* adults as the myofibrils and mitochondria were severely damaged (Figure 4Ag’). In summary, we concluded that mitochondria were shortened in the IFMs with silenced *Polγ*.

### 3.4. Stimulation of Autophagy in the IFMs Harboring PolγRNAi as the Flies Aged

Mitochondrial fission is usually regarded as the stage preceding mitophagy [44]. We investigated whether autophagy occurred in the muscle and whether this took place in the muscle of adults harboring *PolγRNAi*. To quantify mitophagy, we counted anti-Atg8 immunostaining foci overlapping with a mitochondrial marker (anti-ATP5A immunostaining signal) (Figure 5A–C). First, we quantified Atg8 foci, an indicator of autophagy in *Polγ*-depleted muscle. Compared with that in controls (*Mef2^ts^* > *+*, *n* = 22), the Atg8 signal did not increase (*n* ≥ 20) in the IFMs of 5- or 20-d-old adults harboring *PolγRNAi* (*Mef2^ts^* > *PolG2RNAi^1^*, *Mef2^ts^* > *PolG2RNAi^2^*, *Mef2^ts^* > *PolG1RNAi^1^*, and *Mef2^ts^* > *PolG1RNAi^2^*). These differences were not significant for any of the four RNAi types (*p* > 0.05, one-way ANOVA with Bonferroni’s multiple comparisons test) (Figure 5B). With age, however, more Atg8 foci appeared in the IFMs of 36-d-old adults harboring *PolγRNAi* (*Mef2^ts^* > *PolG2RNAi^1^*, *Mef2^ts^* > *PolG2RNAi^2^*, *Mef2^ts^* > *PolG1RNAi^1^*, and *Mef2^ts^* > *PolG1RNAi^2^*) than in the controls (*Mef2^ts^* > *+*, *n* = 22). The differences were significant between every type of *polγRNAi* adults and control adults (*p* < 0.05, one-way ANOVA with Bonferroni’s multiple comparisons test) (Figure 5B). 

We also quantified the pixel size of Atg8 foci that co-localized with the mitochondria in the IFMs of 5-d-old adult muscle harboring *PolγRNAi* (*n* ≥ 20) (Figure 5C). No significant differences in pixel size between 5- and 20-d-old adults harboring *PolγRNAi* and controls were observed (*n* = 22, one-way ANOVA with Bonferroni’s multiple comparisons test) (Figure 5C). In contrast, the mitophagy signal appeared in 36-d-old adults harboring *PolγRNAi* (*n* = 20) more on average than it did in the control at the same age (*Mef2^ts^* > *+*, *n* = 22) (Figure 5C). However, the significant differences were observed only between *Mef2^ts^* > *polG1RNAi^1^* adults and control adults, but not in other RNAi types (*p* > 0.001, one-way ANOVA with Bonferroni’s multiple comparisons test)(Figure 5C). 

To ascertain these results, we performed immunostaining with another antibody that recognized the autophagy adapter protein, Ref(2)P (Appendix A). This adaptor protein is decomposed together with target proteins by autophagy [27]. The Ref(2)P signal did not significantly differ between 5-d-old adult muscles harboring *PolγRNAi* (*Mef2^ts^* > *PolG2RNAi^1^*, *Mef2^ts^* > *PolG2RNAi^2^*, and *Mef2^ts^* > *PolG1RNAi^1^*) and controls (*Mef2^ts^* > *+*) (*n* ≥ 20) (*p* > 0.05, one-way ANOVA with Bonferroni’s multiple comparisons test). Conversely, in the IFMs from 36-d-old adults harboring *PolγRNAi* (*Mef2^ts^* > *PolG2RNAi^1^*, *Mef2^ts^* > *PolG2RNAi^2^*, *Mef2^ts^* > *PolG1RNAi^1^*, and *Mef2^ts^* > *PolG1RNAi^2^*) (*n* ≥ 20), the amount of anti-Ref(2)P foci significantly decreased compared with that in control adults of the same age (*Mef2^ts^* > *+*, *n* = 21) (*p* < 0.0001, one-way ANOVA with Bonferroni’s multiple comparisons test) (Appendix A). These results were consistent with the conclusion that autophagy in the IFMs was stimulated and the targets were soon removed. In summary, these results suggest that both autophagy and mitophagy were partially enhanced to eliminate damaged proteins and abnormal mitochondria with reduced activity in adult muscles.

### 3.5. Continuous Silencing of the Polγ Genes in the Adult Muscle Resulted in Enhanced Apoptosis and Malformation of Muscle Mitochondria and Myofibrils

Suppressed mitochondrial membrane potential is associated with the induction of apoptosis [45]. We investigated whether the effector caspase Dcp1 was activated in the adult muscle, accumulating mtDNA damage (Figure 6A,B). We performed anti-cDcp1 immunostaining of IFMs to detect the activated form of Dcp1 (cDcp1). In young 5-d-old adults harboring *Pol**γRNAi*, the immunostaining signal did not increase significantly compared with that in the control (*Mef2^ts^* > *+*) (Figure 6). In contrast, more anti-cDcp1 foci were detected in IFMs harboring *PolγRNAi* in 20-d-old adults (*Mef2^ts^* > *PolG2RNAi^1^*, *Mef2^ts^* > *PolG2RNAi^2^*, *Mef2^ts^* > *PolG1RNAi^1^*, and *Mef2^ts^* > *PolG1RNAi^2^*) than in controls (*Mef2^ts^* > *+*) at the same age (Figure 6B). Differences in the amount of cDcp1 were significant only in 20-d-old *Mef2^ts^* > *PolG2RNAi^2^* adults (*p* < 0.01, one-way ANOVA with Bonferroni’s multiple comparisons test) but not in other RNAi types (*p* > 0.05) (Figure 6B). Consistently, in 36-d-old adults harboring *PolγRNAi*, more cDcp1 foci were observed than in controls (*Mef2^ts^* > *+*); however, significant differences were not observed in the amount of cDcp1 in every *PolγRNAi* type (*p* > 0.05) (Figure 6B). These results suggest that apoptosis was enhanced in adult muscles through the accumulation of damaged mitochondria owing to *PolγRNAi*.

We examined whether morphological abnormalities in myofibrils and mitochondria were observed in the adult muscle harboring *PolγRNAi*. For this purpose, we carefully observed the IFM of aged adults harboring *PolγRNAi* under a transmission electron microscope (*n* = 5) (Figure 7A–F). In 36-d-old control adult IFMs (*Mef2^ts^* > *+*), no morphological abnormalities were detected in the myofibrils among the 35 electron micrograph fields, except for dense spots that appeared in many mitochondria. These spots were also observed in normal-looking mitochondria, although they appeared less frequently in some myofibrils in the control muscle. In contrast, we frequently observed abnormally distorted Z-lines in the myofibrils (29 out of 69 micrographs) of 36-d-old adults (*Mef2^ts^* > *PolG2RNAi^1^* and *Mef2^ts^* > *PolG1RNAi^1^*) (Figure 7C–E). Additionally, abnormal mitochondria harboring disintegrated cristae (arrow in Figure 7C,D) and those lacking cristae (yellow arrow in Figure 7C) were observed in every IFM examined (33 out of 33 micrographs) in *Mef2^ts^* > *PolG2RNAi^1^* adults. Mitochondria less stained by uranyl acetate appeared abnormally swollen and were also observed in the muscle of aged flies harboring *PolγRNAi* (Figure 7E).

### 3.6. Downregulation of the Polγ Genes in the Muscle Shortened Adult Lifespan and Accelerated Age-Dependent Impairment of Locomotion

As we showed that the silencing of *Polγ* subunit mRNAs resulted in reduced mitochondrial activity in the adult muscle, we performed a lifespan assay in adults harboring muscle-specific *PolG2RNAi* or *PolG1RNAi* (Figure 8A,B). The RNAi flies exhibited a significantly shorter lifespan (*n* > 100) (*Mef2^ts^* > *PolG2RNAi^1^*, *p* = 0.0008; *Mef2^ts^* > *PolG2RNAi^2^*, *Mef2^ts^* > *PolG1RNAi^1^*, and *Mef2^ts^* > *PolG1RNAi^2^*, *p* < 0.0001; log-rank test) than control flies (*Mef2^ts^* > ) (Figure 8A). The median survival time in all flies harboring *PolγRNAi* was shorter than that in controls; it was 43 d after eclosion in the controls, 41 d in *Mef2^ts^* > *PolG2RNAi^1^*, 39 d in *Mef2^ts^* > *PolG2RNAi^2^*, 37 d in *Mef2^ts^* > *PolG1RNAi^1^*, and 35 d in *Mef2^ts^* > *PolG1RNAi^2^*. We also investigated the viability of adults under PQ feeding conditions (Figure 8B). Consistent with the shortening of the adult lifespan, the flies harboring muscle-specific depletion of PolG2 showed a significantly shorter lifespan under 10 mM PQ (*p* < 0.001, log-rank test). Therefore, it is consistent with the shortening of the adult lifespan that the downregulation of PolG2 subunits in adult muscle resulted in a shorter lifespan under PQ-induced oxidative stress.

To examine whether adults harboring muscle-specific *PolγRNAi* would show a loss of locomotor activity, we measured the climbing activity every 5 d from 0 to 35 d after eclosion, until the survival rate reached 50%. The climbing activity of the controls (*Mef2^ts^* > *+*) decreased with age (*n* ≥ 100) (Figure 9). Consistent with the shortened lifespan and higher sensitivity to oxidative stress, the locomotor activities of adults harboring *PolG2RNAi* (*Mef2^ts^* > *PolG2RNAi^1^* and *Mef2^ts^* > *PolG2RNAi^2^*) and *PolG1RNAi* (*Mef2^ts^* > *PolG1RNAi^1^* and *Mef2^ts^* > *PolG1RNAi^2^*) declined even more notably, compared to the control (*Mef2^ts^* > *+*). There were significant differences between every type of *PolgRNAi* adults except *Mef2^ts^* > *PolG2RNAi^2^* and control adults (*p* < 0.0001, two-way ANOVA with Tukey’s multiple comparisons test). There were also significant differences between the time (*p* < 0.0001, two-way ANOVA), and in the group-time interaction (*p* < 0.0001, two-way ANOVA). In summary, the age-dependent decline in locomotion activity became more severe as *Polγ*-depleted adults aged.

Finally, we examined whether *PolγRNAi* influenced circadian rhythm (Appendix A). We used adults harboring *PolG1RNAi* (*Mef2^ts^* > *PolG1RNAi^1^* and *Mef2^ts^* > *PolG1RNAi^2^*), which exhibited a significantly shorter lifespan. The locomotion efficiency of *PolG1*-depleted adults was measured for 24 h (in a 12/12-h light/dark cycle) until the survival rate reached 50% (*n* ≥ 20). Here, 5-, 15-, and 35-d-old adults were compared in terms of the highest levels of daily activity. In control adults (*Mef2^ts^* > *+*), the highest level gradually decreased with aging (Appendix A). In contrast, the extent of the decline was more remarkable in 5-d-old adults of the same age harboring *Mef2^ts^* > *PolG1RNAi^1^* (68% of the activity of control adults (*Mef2^ts^* > *+*)) and *Mef2^ts^* > *PolG1RNAi^2^
*(84% of the activity of control adults (*Mef2^ts^* > *+*)). A similar trend was observed for the 15- (55% and 64% of the activity of controls) and 35-d-old (68% and 84% of the activity of controls) adults (Appendix A). The locomotor activity of 35-d-old adults harboring *PolG1RNAi* increased during the daytime (between 11:00–19:00 h), whereas the control flies and younger RNAi flies were less active during this period. In summary, these results indicate that muscle-specific silencing of *Polγ* accelerated the age-dependent decline in locomotor activity. *Polγ* silencing resulted in a disturbance of the regular circadian rhythm of aged adults, who carried fewer active mitochondria with accumulated mtDNA damage.

## 4. Discussion

### 4.1. Accumulation of Mitochondrial DNA Damage and Promotion of Mitochondrial Dysfunction in Polγ-Silenced Adult Muscle

We showed that the muscle-specific silencing of the subunits of DNA Polγ, the sole DNA polymerase in mitochondria, resulted in the partial accumulation of oxidative damage in mtDNA and a reduction in its copy number. This resulted in the production of abnormal mitochondria with reduced membrane potential required for ATP production in the adult muscle. However, the severity of mitochondrial damage in adult muscles harboring *PolG2RNAi* or *PolG1RNAi* was not always correlated with mRNA levels. For example, there were no significant differences in silencing efficiency between *PolG2RNAi* and *PolG1RNAi* at 20 d of age. Nevertheless, mtDNA repair in 10-d-old adults fed PQ and mtDNA replication in 36-d-old adults were rather more severely inhibited in adults harboring *PolG2RNAi* than in those harboring *PolG1RNAi*. PolG1 is broken down by the protease LONP1, and PolG2 protects the catalytic subunit from this degradation. PolG2 knockdown has been shown to reduce PolG1 protein levels in mice [46]. The amount of Polγ holoenzyme depends on the *PolG2* mRNA levels if there is no regulatory mechanism during the translational process. We demonstrated that *PolG2* mRNA levels decreased with aging, whereas *PolG1* mRNA remained at a constant level during normal aging. Considering that the amount of Polγ is determined by the amount of PolG2 in mice [46], the amount of *Drosophila* Polγ complex decreases with aging. As the *PolG2* mRNA for Polγ accessory subunits progressively declines during normal aging, the amount of Polγ holoenzyme decreases accordingly. The amount of the Polγ complex may depend mainly on the *PolG2* mRNA levels during the normal aging process. Considering this together with the aspect of the protection of PolG1 from proteases by PolG2, we propose that downregulation of the Polγ complex will be more accelerated in adults harboring *PolG2RNAi* than in those harboring *PolG1RNAi*. Hence, there was a trend toward appearance of more severe phenotypes in *PolG2* RNAi than that in *PolG1*RNAi.

mtDNA damage was higher in the IFMs harboring *Mef2^ts^* > *PolG2RNAi* and *Mef2^ts^* > *PolG1RNAi* than in controls. Aberrations in mtDNA are frequently detected in tissues that produce ROS at higher levels, followed by mtDNA damage [47]. Mutations in mtDNA increase with age in the skeletal muscle of aged people [48], which may result from the age-related decline in the mtDNA repair and replication functions [49]. Mutations in the mitochondrial genome are associated with a range of human diseases and have been implicated as a driving force behind the aging process [50]. A wide range of mtDNA sequencing results have revealed that mtDNA mutations accumulate in the liver of mice expressing a proofreading-deficient mtDNA polymerase. The mutator mice had 10-fold higher levels of point mutations, but not large deletions, accumulated more with age than normal siblings [51]. Premature aging occurs in mtDNA mutator mice. Consistently, we also observed that mtDNA damage results in the production of abnormal mitochondria with reduced membrane potential required for ATP production in the adult muscle. In addition to the stimulation of mitochondrial fission, autophagy, were significantly increased in the muscle with the mtDNA mutator. There was a trend toward increase in apoptosis in the muscle. Eventually, malformation of both myofibrils and mitochondria, reminiscent of sarcopenia in humans, was frequently observed in the muscle. Our data are consistent with previous data from mice carrying the mtDNA mutator [46]. However, no increase in point mutations or deletions was observed in control mice during normal aging. These results are inconsistent with the hypothesis that the accumulation of mtDNA mutations contributes to aging [51]. Another study on aged human brains did not find significant changes in mtDNA related to oxidative damage [12]. In contrast, we demonstrated that the expression of the PolG2 subunit decreases as flies age, and that of PolG1 may also decrease in association with the age-dependent decline of PolG2. It is necessary to investigate this further by determining the mtDNA sequences of the flies harboring *Pol**γRNAi* and test the hypothesis that mtDNA damage progressively accumulates in the adult muscle of *pol**γRNAi* or normal flies. Establishing whether mtDNA damage contributes to the progression of aging-related phenotypes and addressing whether un-repaired mtDNA accumulates owing to ROS production in mitochondria with reduced activity are crucial. 

### 4.2. Acceleration of Mitochondrial Fission, Followed by Mitophagy in Adult Muscle Carrying Abnormal Mitochondria with Accumulating DNA Damage

Here, we found that the silencing of Polγ subunits facilitated mtDNA damage accumulation, and the production of abnormal mitochondria showing reduced activity increased in the adult muscle. Consequently, mitochondrial fission was significantly enhanced. Mitophagy was partially enhanced in the muscle of aged *mef2^ts^* > *Polγ*RNAi flies. These results imply that mitochondrial fission was promoted and, thereafter, mitophagy was stimulated in the muscle harboring *PolγRNAi*. Consistent with this idea, several in vitro studies have reported that dysfunctional mitochondria generated under increased ROS production are selectively eliminated by mitophagy [52,53]. Another study showed that inhibiting mitochondrial fission reduces mitophagy induction, supporting this idea. Mitochondrial fission usually occurs before mitophagy [54] and is enhanced in mammalian muscle cells with excessive ROS accumulation [45,55]. In other words, the fission of damaged mitochondria under excessive ROS is stimulated; subsequently, only the resultant smaller mitochondria with damaged cellular components are removed by mitophagy. Mitophagy and autophagy play indispensable roles in maintaining cell homeostasis and overexpression of factors involved in either process resulted in a prolonged lifespan of many model organisms [56,57,58,59]. However, there is conflicting evidence that excessive autophagy enhances cell death in *Drosophila* and *C. elegans* [60,61]. Therefore, it is important to further examine whether excessive autophagy affects muscle activity and locomotor activity with aging; the *Drosophila* system used here provides a good tool to address this issue.

### 4.3. Age-Dependent Impairment of Mitochondrial Activity, and Aging-Related Phenotypes and Behavior in the Adult Muscle

Oxidative DNA damage of mitochondria is considered the major cause of apoptosis and sarcopenia observed in advanced muscle aging in mammals [62,63,64,65]. Here, we found that the age-dependent decline in locomotor activity was accelerated in flies harboring *PolγRNAi*. We suspected that locomotor impairment in these flies was related to muscle dysfunction in aged flies. As *Drosophila* is a small model organism, it is difficult to quantify or estimate the amount or loss of muscle mass during aging [66]. Therefore, instead of quantifying muscle mass, we observed myofibrils by TEM and examined whether there were any structural abnormalities that suggested the loss of cellular components in muscle cells. We observed abnormal myofibrils, such as distorted Z-lines, in the aged adults harboring *PolγRNAi*. A previous study of muscle dystrophy models in *Drosophila* showed similar Z-line abnormalities in aged flies homozygous for mutations in the *tw* gene encoding mannosyltransferase and that apoptosis is enhanced simultaneously [67]. Therefore, we hypothesized that damaged myofibrils in *Polγ*-silenced adults are generated in association with apoptosis. Especially in *PolG2RNAi^2^* adults, the apoptosis signal detected by anti-cDcp1 immunostaining showed an increase from the early adult stage. Additionally, remarkable damage to the myofibrils and inner and outer mitochondrial membranes was observed in the muscle cells of *Polγ*-silenced adults. These phenotypes supported the idea that apoptosis is continuously stimulated from the early adult stage, and this enhanced apoptosis eventually leads to non-negligible abnormalities in myofibril and mitochondrial structures in adult muscles. In mammalian cells undergoing sarcopenia, apoptosis and muscle atrophy are enhanced after mitochondrial fission [68,69,70]. Therefore, the accumulation of damaged mitochondria owing to Polγ-downregulation in adult muscle may stimulate mitochondrial fission, resulting in the induction of apoptosis in muscle cells. 

Next, we discuss how ROS and oxidative DNA damage in mtDNA ultimately affect lifespan. These adults exhibited a shortened lifespan and accelerated age-dependent impairment of locomotor activity. Most ROS are generated during oxidative phosphorylation in the mitochondrial electron transport chain [4]. Although moderate levels of ROS have been reported to prolong lifespan [71], higher ROS production levels are thought to cause mtDNA mutations, membrane potential loss, mitochondrial dysfunction, and cell death. Ultimately, they accelerate the progression of aging [72]. We also obtained evidence suggesting that the mitochondrial membrane potential was reduced and, consequently, ATP production deteriorated in adult muscle harboring *Pol**γRNAi*. Apoptosis is promoted and, eventually, the adult lifespan is shortened. This finding is consistent with a previously proposed model that mutations accumulated in mitochondrial DNA during the aging process may be causally related to the decreased physiological responses of senescent organisms [72]. Eventually, the muscle phenotype may have led to an age-dependent decline in locomotor activity. In addition to locomotor activity, it is important to investigate whether other aging-related phenotypes are accelerated in adults that harbor muscle-specific *PolγRNAi*. Such processes include proteostasis impairment, which leads to ubiquitinated protein aggregate formation. We propose to explore these aspects in our future studies. 

We observed that muscle-specific *PolG2RNAi* disturbed the circadian rhythm. Similar symptoms are sometimes observed in elderly people with sarcopenia in their muscles [73,74]. However, it is still necessary to more carefully examine the relationship between muscle-aging phenotypes originating from mitochondrial dysfunction and disturbed circadian rhythms, although one study has described that circadian rhythm disruption influences sarcopenia through skeletal muscle dysfunction and vascular disorders during aging [74]. It would be interesting to perform studies in future to clarify the relationship between myokines secreted from muscle and circadian clock regulators [75,76]. The evidence derived from this study is consistent with the hypothesis that the dysfunction of organelles originating from the accumulation of mtDNA damage can contribute to the aging of *Drosophila* muscle.

## Figures and Tables

**Figure 1 biomolecules-12-01105-f001:**
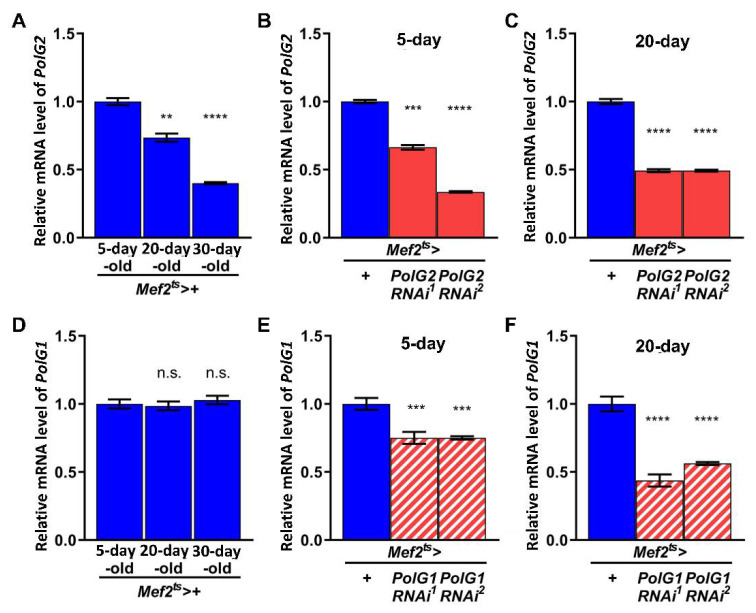
Expression and double-stranded RNA-mediated silencing of mRNAs encoding Polγ subunits in *Drosophila* adults with aging. (**A**–**F**) Relative mRNA levels of *PolG2* (**A**–**C**) and *PolG1* (**D**–**F**) in 5-, 20-, and 36-day-old-flies. (**A**,**D**) mRNA levels of two genes quantified by qRT-PCR. The average levels in control flies at the same age (*Mef2^ts^* > *+*; the F1 progenies derived from a cross between the *Mef2^ts^*-Gal4 stock and a standard wild stock, Canton S) are presented as 1.0. Note that the mRNA levels of *PolG2* mRNA decreased with age, whereas *PolG1* mRNA levels remained constant. (**B**,**C**,**E**,**F**) mRNA levels of flies with muscle-specific expression of double-stranded RNA against *PolG2* or *PolG1* mRNA were quantified by qRT-PCR (*Mef2^ts^* > *PolG2RNAi* and *Mef2^ts^* > *PolG1RNAi*). The average levels in control flies at the same age (*Mef2^ts^* > *+*) are presented as 1.0. Each mRNA level was reduced to 34–75% of the controls. Error bar; s.e.m. ((**A**,**D**); *n* = 3 (triplicates), (**B**,**C**,**E**,**F**); *n* = 9 (three biological triplicates), ** *p* < 0.01, *** *p* < 0.001, **** *p* < 0.0001, n.s. not significant, one-way ANOVA with Bonferroni’s multiple comparisons test).

**Figure 2 biomolecules-12-01105-f002:**
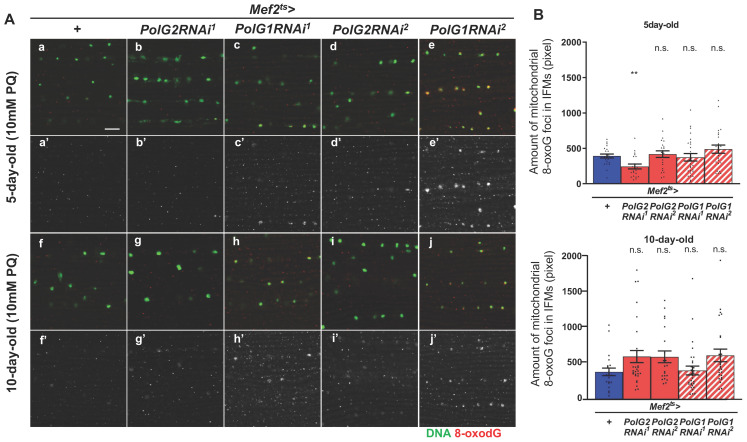
Quantification of mitochondrial DNA (mtDNA) containing 8-oxo-deoxyguanine (8-oxodG) in the indirect flight muscles (IFMs) of the *PolG1RNAi* or *PolG2RNAi* adults fed paraquat (PQ). (**A**) Immunostaining of the IFMs collected from 5- and 10-day-old adults fed fly food supplemented with 10 mM PQ revealed with an anti-8-oxodG antibody (red) and PicoGreen for DNA visualization (green in **a**–**j**, white in **a’**–**j’**). Scale bar: 10 μm. *PolG2RNAi* flies (*Mef2^ts^* > *PolG2RNAi^1^* and *Mef2^ts^* > *PolG2RNAi^2^*), *PolG1RNAi* flies (*Mef2^ts^* > *PolG1RNAi^1^* and *Mef2^ts^* > *PolG1RNAi^2^*), and control flies (*Mef2^ts^* > *+*) were examined. (**B**) Average number of 8-oxodG foci per confocal optical field (4.0 × 10^−2^ mm^2^) in the muscles of 5- and 10-day-old adults harboring *PolγRNAi* (*Mef2^ts^* > *PolG2RNAi^1^* (*n* = 21 for 5-day-old, *n* = 30 for 10-day-old), *Mef2^ts^* > *PolG2RNAi^2^* (*n* = 22, 21), *Mef2^ts^* > *PolG1RNAi^2^* (*n* = 26, 30), and *Mef2^ts^* > *PolG1RNAi^1^* (*n* = 24, 25)) and controls (*Mef2^ts^* > *+*). (** *p* < 0.01, n.s.; not significant. One-way ANOVA with Bonferroni’s multiple comparisons test).

**Figure 3 biomolecules-12-01105-f003:**
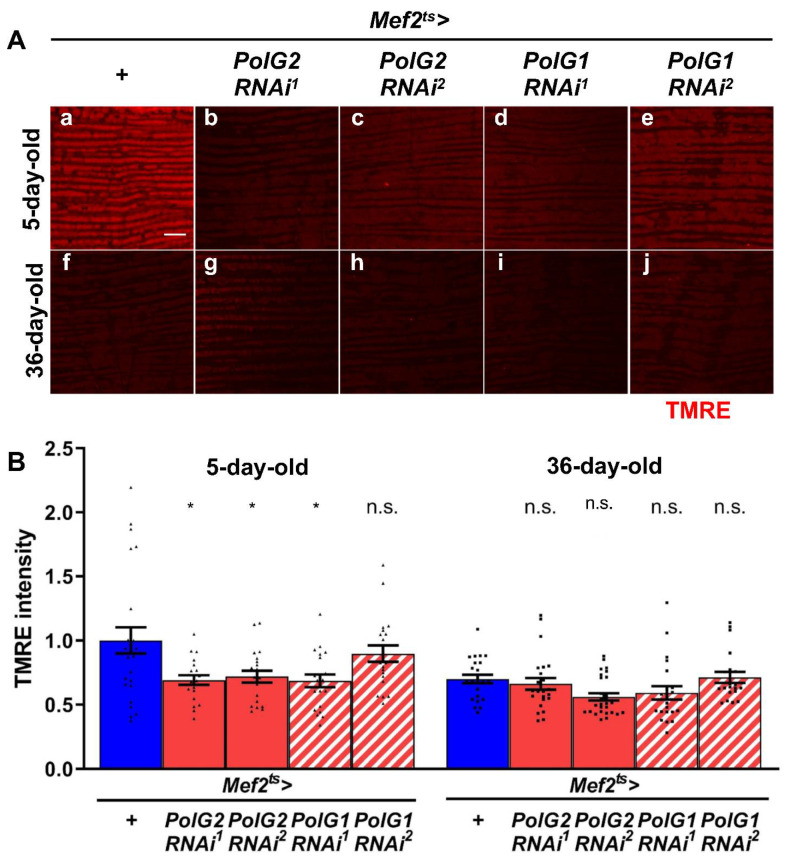
Measurement of mitochondrial membrane potential in the adult muscles harboring *PolG1RNAi* and *PolG2**RNAi*. (**A**) Tetramethylrhodamine ethyl ester (TMRE) staining of the IFMs of 5- and 36-day-old control adults and adults harboring muscle-specific *PolG2RNAi* and *PolG1RNAi* (red). Scale bar: 10 μm. (**B**) Quantification of TMRE intensity per confocal field (4.0 × 10^−2^ mm^2^) in the IFMs of 5- and 36-day-old adult flies harboring muscle-specific *polG2RNAi* (*Mef2^ts^* > *PolG2RNAi^1^* (*n* = 20 for 5-day-old, *n* = 24 for 36-day-old), *Mef2^ts^* > *PolG2RNAi^2^* (*n* = 20, 28) and *polG1RNAi* (*Mef2^ts^* > *PolG1RNAi^1^* (*n* = 20, 22) and *Mef2^ts^* > *PolG1RNAi^2^* (*n* = 20, 20)), normalized to the control adults at the same age (*Mef2^ts^* > *+* (*n* = 25, 24)). (* *p* < 0.05, n.s.; not significant, one-way ANOVA with Bonferroni’s multiple comparisons test).

**Figure 4 biomolecules-12-01105-f004:**
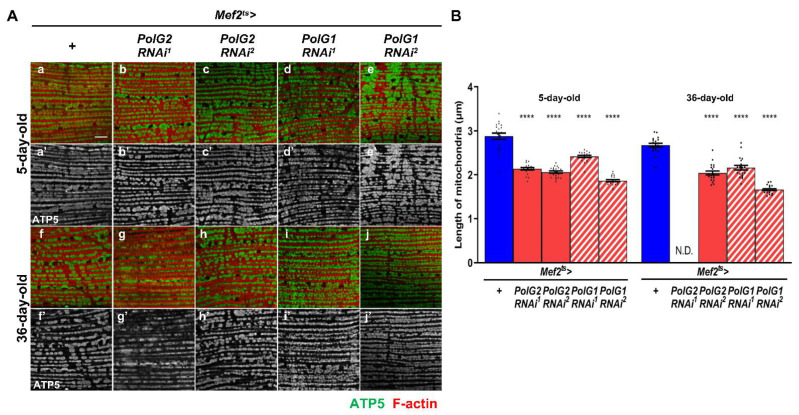
Observation of mitochondria by confocal microscopy and quantification of the mitochondrial length in the adult IFMs harboring muscle-specific *PolG1RNAi* and *PolG2**RNAi*. (**A**) Anti-ATP5A immunostaining of the IFMs collected from 5- and 36-day-old adults to visualize entire mitochondria of control adults (*Mef2^ts^* > *+*) and adults harboring muscle-specific *PolG1RNAi* (*Mef2^ts^* > *PolG2RNAi^1^* and *Mef2^ts^* > *PolG2RNAi^2^*) and *PolG1**RNAi* (*Mef2^ts^* > *PolG1RNAi^1^* and *Mef2^ts^* > *PolG1RNAi^2^*). Anti-ATP5A antibody staining (green in **a**–**j**, white in **a’**–**j’**) and Alexa488-phalloidin staining for F-actin (red). Scale bar: 10 μm. (**B**) Quantification of muscular mitochondrial length in control adults (*Mef2^ts^* > *+*) and adults harboring muscle-specific *PolG2RNAi* and *PolG1RNAi* (*Mef2^ts^* > *PolG2RNAi^1^*, *Mef2^ts^* > *PolG2RNAi^2^*, *Mef2^ts^* > *PolG1RNAi^1^*, and *Mef2^ts^* > *PolG1RNAi^2^*). 20 adult thoraxes at 5-day and 36-day old were examined in every genotype. N.D.; 36-day-old *Mef2^ts^* > *PolG2RNAi^1^* flies contained mitochondria in which the mitochondrial membrane was too distorted to measure the length. (*n* = 20, **** *p* < 0.0001, one-way ANOVA with Bonferroni’s multiple comparisons test). Error bar; s.e.m.

**Figure 5 biomolecules-12-01105-f005:**
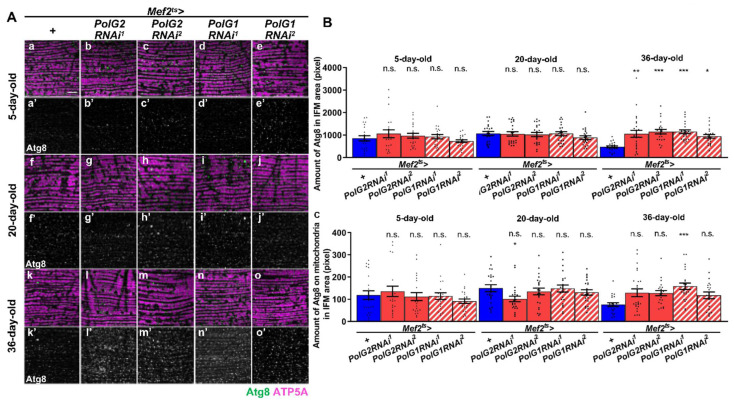
Immunostaining and quantification of autophagy marker protein, Atg8 in IFMs of adults harboring *Polγ**RNAi*. (**A**) Immunostaining of IFMs from 5-, 20-, and 36-day-old control adults or 5-, 20-, and 36-day-old adults harboring *Polγ**RNAi* with anti-Atg8 (green in **a**–**o**, white in **a’**–**o’**) and anti-ATP5A antibodies to visualize mitochondria (magenta). Scale bar: 10 μm. (**B**) Quantification of the Atg8 foci in the IFM cells of 5-, 20- and 30-day-old control flies (*Mef2^ts^* > *+* (*n* = 22 for 5-day-old, 20-day-old, and 36-day old)) and flies harboring muscle-specific *Pol G2RNAi* and *PolG1RNAi* (*Mef2^ts^* > *PolG2RNAi^1^* (*n* = 21, 22, 25), *Mef2^ts^* > *PolG2RNAi^2^* (*n* = 22, 22, 25), *Mef2^ts^* > *PolG1RNAi^1^* (*n* = 22, 22, 23), and *Mef2^ts^* > *PolG1RNAi^2^* (*n* = 22, 22, 21). (**C**) Quantification of the Atg8 foci co-localized with mitochondria in the IFMs of 5-, 20-, and 36-day-old control flies (*Mef2^ts^* > *+ n* = 22 for 5-day-old and 20-day-old, *n* = 23 for 36-day old)) and flies harboring *PolG2RNAi* and *PolG1RNAi* (*Mef2^ts^* > *PolG2RNAi^1^* (*n* = 21, 23, 26), *Mef2^ts^* > *PolG2RNAi^2^* (*n* = 23 for all three age groups), *Mef2^ts^* > *PolG1RNAi^1^* (*n* = 23, 23, 24), and *Mef2^ts^* > *PolG1RNAi^2^* (*n* = 23 for all three age groups). (* *p* < 0.05, ** *p* < 0.01, *** *p* < 0.001, n.s.; not significant, one-way ANOVA with Bonferroni’s multiple comparisons test). Error bars; s.e.m.

**Figure 6 biomolecules-12-01105-f006:**
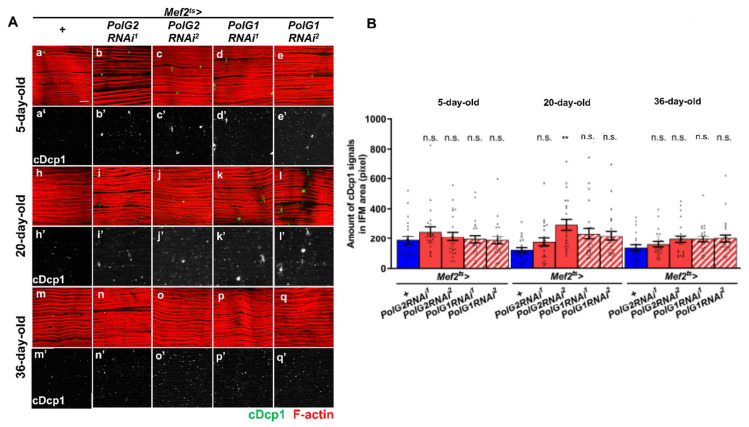
Immunostaining of adult IFMs harboring *pol**γ**RNAi* with anti-cDcp1 antibody and quantification of the foci representing activation of the effector caspase. (**A**) Immunostaining of IFMs collected from 5-, 20-, and 36-day-old control adults (*Mef2^ts^* > *+*) and adults harboring muscle-specific *PolG2RNAi* and *PolG1RNAi* (*Mef2^ts^* > *PolG2RNAi^1^*, *Mef2^ts^* > *PolG2RNAi^2^*, *Mef2^ts^* > *PolG1RNAi^1^*, and *Mef2^ts^* > *PolG1RNAi^2^*) with anti-cDcp1 antibody (green in **a**–**q**, white in **a’**–**q’**). Myofibrils of the IFMs are visualized by phalloidin staining (red). Scale bar: 10 μm. (**B**) Quantification of the cDcp1 foci in the IFMs from 5-, 20-, and 36-day-old control adults (*Mef2^ts^* > *+* (*n* = 22 for 5-day-old, *n* = 23 for 20-day-old, and *n* = 22 for 36-day old)) and adults harboring muscle-specific *PolG2RNAi* and *PolG1RNAi* (*Mef2^ts^* > *PolG2RNAi^1^* (*n* = 22, 23, 24) *Mef2^ts^* > *PolG2RNAi^2^* (*n*= 23 for all three age groups), *Mef2^ts^* > *PolG1RNAi^1^* (*n* = 22, 23, 23) and *Mef2^ts^* > *PolG1RNAi^2 2^
*(*n* = 23 for all three age groups). (** *p* < 0.01, n.s.; not significant, one-way ANOVA with Bonferroni’s multiple comparisons test). Error bars; s.e.m.

**Figure 7 biomolecules-12-01105-f007:**
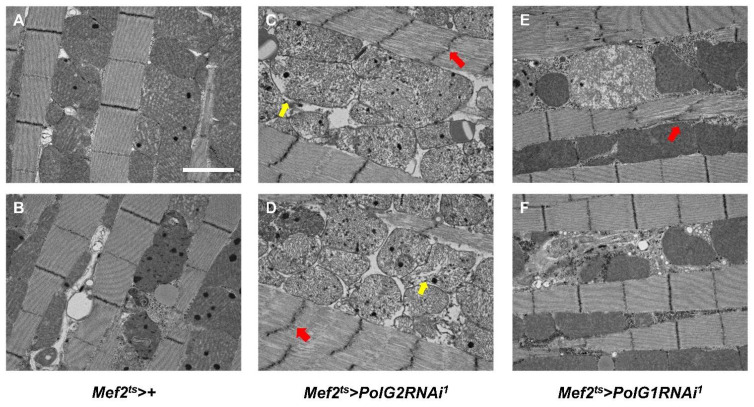
Transmission electron microscopy observation of indirect flight muscles from aged flies harboring continuous muscle-specific *Polγ**RNAi*. (**A**–**F**) Observation of the myofibrils and mitochondria in the indirect flight muscles (IFMs) of 36-day-old control adults by transmission electron microscopy (TEM). (**A**,**B**) IFM cells from control adults (*Mef2^ts^* > *+*) and (**C**,**D**) those from adults harboring muscle-specific *PolG2RNAi* (*Mef2^ts^* > *PolG2RNAi^1^*) (**C**) and *PolG1RNAi* (*Mef2^ts^* > *PolG1RNAi^1^*) (**D**) were observed. Distorted Z-lines in the myofibrils of the *Polγ**RNAi* IFMs were frequently observed (red arrows) in *Mef2^ts^* > *PolG2RNAi^1^* (**C**,**D**) and *Mef2^ts^* > *PolG1RNAi^1^* (**E**). Disintegrated cristae (yellow arrows in (**C**,**D**)) were more frequently observed in the IFMs of *Polγ**RNAi* adults (*Mef2^ts^* > *PolG1RNAi^1^*). Note that the ultra-thin section shown in (**B**) is not parallel to the mitochondrial lines. Scale bar: 2 μm.

**Figure 8 biomolecules-12-01105-f008:**
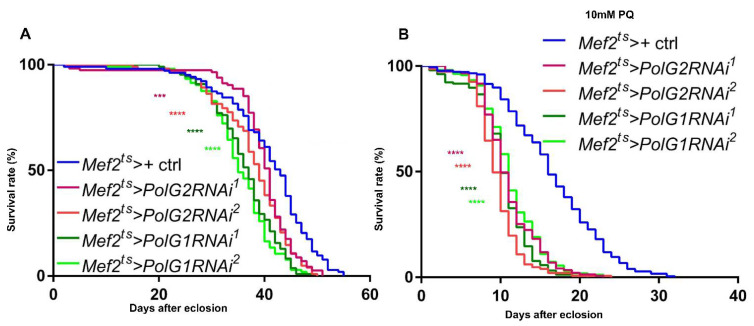
Lifespan curves of adults harboring muscle-specific *PolG1RNAi* and *PolG2**RNAi* and survival curves of adults fed PQ. (**A**,**B**) Lifespan curves of control adults (*Mef2^ts^* > *+*) and *PolγRNAi* adults (*Mef2^ts^* > *PolG2RNAi^1^*, *Mef2^ts^* > *PolG2RNAi^2^*, *Mef2^ts^* > *PolG1RNAi^1^*, and *Mef2^ts^* > *PolG1RNAi^2^*) fed on the instant fly food (**A**) or the food supplemented with 10 mM PQ (**B**). The flies collected within 1 day after eclosion were used for (**A**) (*n* = 103 control adults, *n* = 114 *Mef2^ts^* > *PolG2RNAi^1^* adults, *n* = 102 *Mef2^ts^* > *PolG2RNAi^2^*, *n* = 109 *Mef2^ts^* > *PolG1RNAi^1^*, and *n* = 104 *Mef2^ts^* > *PolG1RNAi^2^*) and used for (**B**) (*n* = 177 control adults, *n* = 143 *Mef2^ts^* > *PolG2RNAi^1^* adults, *n* = 147 *Mef2^ts^* > *PolG2RNAi^2^*, *n* = 156 *Mef2^ts^* > *PolG1RNAi^1^*, and *n* = 163 *Mef2^ts^* > *PolG1RNAi^2^*). More than 5 trials with the total number of adults described were performed in every assay type. The survival rate of the flies was examined every 24 h (*** *p* < 0.001, **** *p* < 0.0001, log-rank test).

**Figure 9 biomolecules-12-01105-f009:**
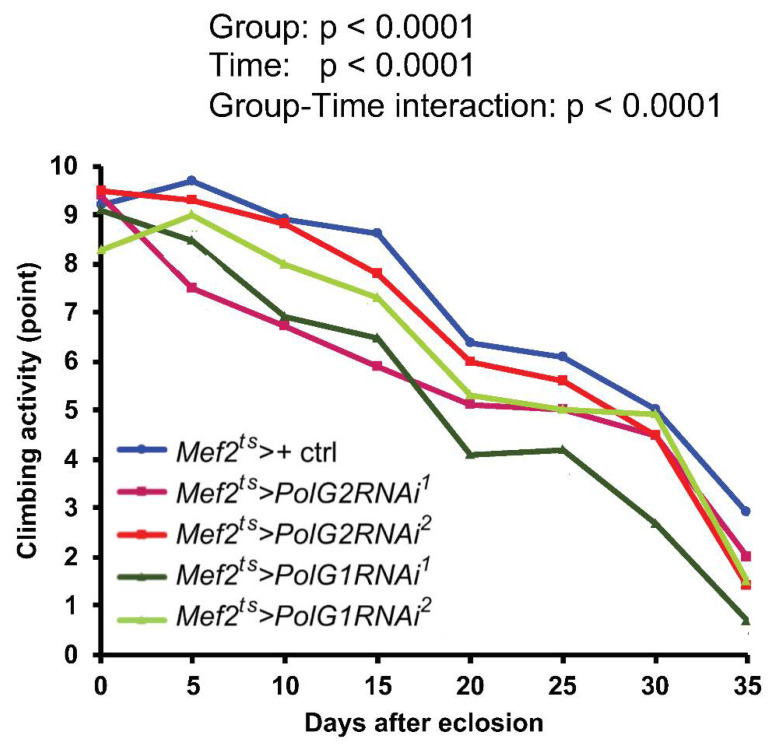
Climbing assay of adults with muscle-specific *PolG1* or *PolG2 RNAi*. Climbing assay to quantify the locomotion activity of adults harboring muscle-specific silencing of mRNAs encoding Polγ subunits (*Mef2^ts^* > *PolG2RNAi^1^*, *Mef2^ts^* > *PolG2RNAi^2^*, *Mef2^ts^* > *PolG1RNAi^1^*, and *Mef2^ts^* > *PolG1RNAi^2^*). The newly eclosed flies were collected within 1 day after eclosion and fed instant fly food. Their climbing activity was examined every 5 d until 35 d after eclosion (*n* = 102 to 145 flies from seven to ten repeated assays). The points on the y-axis show the mean climbing scores representing the locomotor activity of flies. *PolγRNAi* adults showed significantly lower activity than controls (*Mef2^ts^* > *+*) (two-way ANOVA with Tukey’s multiple comparisons test). There were significant differences in the genotypes (*p* < 0.0001); *p* < 0.0001 for a comparison between *PolG2RNAi^1^* and control, between *PolG1RNAi^1^* and control, between *PolG1RNAi^2^* and control. *p* = 0.4085 (not significant) between *PolG2RNAi^2^* and control. There were also significant differences in the time (*p* < 0.0001), and in the genotype-time interaction (*p* < 0.0001).

## Data Availability

The study did not report any data.

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
