# Peer review of "Downregulating Mitochondrial DNA Polymerase γ in the Muscle Stimulated Autophagy, Apoptosis, and Muscle Aging-Related Phenotypes in Drosophila Adults"

_biomolecules, 2022, doi:10.3390/biom12081105_

Round 1

Reviewer 1 Report

Biomolecules review

Manuscript uses RNAi to knock down mitochondrial DNA polymerase in adult muscle and does a nice variety of phenotypic assessments, including cell biology, physiology and molecular biology.  Data is generally well presented and solid. I have a few concerns about statistical methods and interpretations that should be addressed to improve the manuscript.

Major:

1. Stats issues: Authors consistently compare multiple RNAi constructs individually with t-tests, when it would be more appropriate to use an ANOVA with post-hoc corrections for multiple testing (such as a Bonferroni or Tukey test). This applies to most of the data figures. 

The longitudinal climbing data is very nice and I complement the authors for doing the experiment this way, since it provides important information about the slope of change that could not be gained from a cohort strategy. A t-test at individual data points does not take advantage of the longitudinal approach, however. A genotype-by-age analysis would allow conclusions about whether the RNAis are changing the age-related slope, which would be a more powerful and appropriate analysis using the data that is already there.

In the longevity experiment, it is not clear what it means when authors say RNAi causes “high-sensitivity to PQ”.  Are they comparing the same line with or without PQ?  Or comparing all the lines with PQ?  If they want to compare whether RNAi increases sensitivity, they should do a genotype-by-treatment analysis. If they are just comparing longevity on PQ between the genotypes, they should just say “live shorter under PQ treatment”.

2. Authors should do some additional copy editing to make some of the sentences more clear. For example, description of the RNAi experiments in Figure 1 are difficult for the reader to understand. There are a few other places where the sentences could be simplified also.

3. Should be clarified what mef2>+ means. I think this means the driver line has been outcrossed to the attp40 background? But authors should state this directly, perhaps in the first figure legend.

4. In the results for the ATP quantification, I do not feel it is appropriate to say that the results were not significant, but use them in an interpretation anyway. Non-significant changes should not be used as a supporting argument.

5. Early in the manuscript, authors say they are proving the Mitochondrial theory of aging. I’m not sure I agree with that. These results (and those from others) are establishing that mitochondrial dysfunction causes symptoms that are similar to aging.  The later statement the authors make that mitochondrial mutations “contribute to progression of aging in muscle” is much better to me. 

6. In Figure 5B, it doesn’t look to me that Atg8 is accumulating in mutants as the authors say. It looks like the control level is going down at higher ages instead. Does this change the interpretation?

7. I didn’t understand why authors drew attention to lipid droplets in the EM data. The number of lipid droplets observed was quite low in RNAi flies and controls, so I didn’t follow why the authors were so interested in that. Could either be clarified or removed.

8. N-values should be clarified for several experiments. Does the n-value mean the number of biological replicates or the number of flies?

Minor:

1. “depicted” RNA interference should be “experienced” RNA interference, I think?

2. sentence from lines 398-400 about copy number should be moved to the previous section where copy number is being discussed

3. on line 469, it says “autophagy occurred in the mitochondria” I think this gives a mistaken impression. Maybe should say “autophagy occurred in muscle”

4. on line 660, “alleviated” should be “accelerated” I think

Author Response

1.Stats issues: Authors consistently compare multiple RNAi constructs individually with t-tests, when it would be more appropriate to use an ANOVA with post-hoc corrections for multiple testing (such as a Bonferroni or Tukey test). This applies to most of the data figures. 

According to the reviewer’s request, we redid statistical analysis using One or Two-ANOVA with posthoc corrections for multiple testing (a Bonferroni or Tukey test) in all figures except Fig. 7 and Fig. 8, instead of a t-test. We added the following sentence in the statistical analysis of Materials and Method (line 285); “For the comparisons of the two groups, we used the Student’s t-test. One-way ANOVA followed by Bonferroni post-hoc test was applied to assess the differences in more than two groups. Two-way ANOVA followed by Tukey post-hoc was performed to compare the mean differences between groups that were split into two independent variables.” When using a statistical method with a more stringent significance test has eliminated the significant difference, we changed the conclusion assertion in Results and Discussion to a softened and milder wording.

The longitudinal climbing data is very nice and I complement the authors for doing the experiment this way, since it provides important information about the slope of change that could not be gained from a cohort strategy. A t-test at individual data points does not take advantage of the longitudinal approach, however. A genotype-by-age analysis would allow conclusions about whether the RNAis are changing the age-related slope, which would be a more powerful and appropriate analysis using the data that is already there.

We appreciate the reviewer’s comment which encouraged us to continue to promote our research using Drosophila in this direction. We replaced the statistical significance letters with the letters based on two-way ANOVA with Tukey’s multiple comparisons test performed according to the reviewer’s comment in Fig. 9.

In the longevity experiment, it is not clear what it means when authors say RNAi causes “high-sensitivity to PQ”.  Are they comparing the same line with or without PQ?  Or comparing all the lines with PQ?  If they want to compare whether RNAi increases sensitivity, they should do a genotype-by-treatment analysis. If they are just comparing longevity on PQ between the genotypes, they should just say “live shorter under PQ treatment”.

According to the reviewer’s comment, we revised the sentences (used to be on line 522-527) as requested. “the flies harboring muscle specific depletion of PolG2 showed a significantly shorter lifespan under 10 mM PQ (p < 0.001, log-rank test)” (lines 481-483).

2.Authors should do some additional copy editing to make some of the sentences more clear. For example, description of the RNAi experiments in Figure 1 are difficult for the reader to understand. There are a few other places where the sentences could be simplified also.

We asked the professional English proofreader (Editage Co.), a native English speaker, to check our English sentences. According to the proofreader’s suggestion, we revised several sentences in the manuscript to make the manuscript more readable to read. Every edit change made in the manuscript is shown as Track Changes in Word.

3.Should be clarified what mef2>+ means. I think this means the driver line has been outcrossed to the attp40 background? But authors should state this directly, perhaps in the first figure legend.

According to the reviewer’s request, we added a brief explanation about what the Mef2ts>+ means in Figure legend 1 (line 861-862) as follows; Mef2ts>+; the F1 progenies derived from a cross between a Mef2ts-Gal4 stock and a standard wild-type, Canton S.

4,In the results for the ATP quantification, I do not feel it is appropriate to say that the results were not significant, but use them in an interpretation anyway. Non-significant changes should not be used as a supporting argument.

We agreed with the reviewer’s criticism that the ATP levels did not decrease significantly and that we should not use the non-significant results as a supporting argument. Therefore, we first removed the final sentence describing reduced mitochondrial activity in adults harboring the Polg depletion according to the reviewer’s comment (used to be lines 409−411). Second, we also removed the phrase “and ATP levels” in the first part of the discussion (line 5ï¼’ï¼’, which used to be in line 560). Third, we revised another sentence (line 596) as follows; “Consistently, we also showed that mtDNA damage results in the production of abnormal mitochondria with reduced membrane potential required for ATP production in the adult muscle. ” (inserted the phrase underlined, line 522).

5.Early in the manuscript, authors say they are proving the Mitochondrial theory of aging. I’m not sure I agree with that. These results (and those from others) are establishing that mitochondrial dysfunction causes symptoms that are similar to aging.  The later statement the authors make that mitochondrial mutations “contribute to progression of aging in muscle” is much better to me.  

We can understand the reviewer’s criticism. We removed the sentence (used to be in lines 51-52).

6.In Figure 5B, it doesn’t look to me that Atg8 is accumulating in mutants as the authors say. It looks like the control level is going down at higher ages instead. Does this change the interpretation?

We understand the reviewer’s criticism that the Atg8 foci that appeared in adult IFMs harboring PolG depletion at every age do not appear to be significantly increased. By contrast, as we showed in Fig. S3, the foci containing Ref2(2)P, known to be removed as the autophagy stimulates, were significantly decreased. These observations are consistent with the interpretation that autophagy was enhanced to eliminate damaged proteins. However, based on the revised statistical analysis, we revised the sentences describing the alteration of the Atg8 foci (lines 410-413, 431-433)

7.I didn’t understand why authors drew attention to lipid droplets in the EM data. The number of lipid droplets observed was quite low in RNAi flies and controls, so I didn’t follow why the authors were so interested in that. Could either be clarified or removed.

According to the reviewer’s request, we removed three sentences describing lipid droplets (used to be in previous lines 495-498), red arrowheads in Figure 7, and relevant figure legends (used to be in previous lines 950-951).

8.N-values should be clarified for several experiments. Does the n-value mean the number of biological replicates or the number of flies?

According to the reviewers’ request, we added N-value in each figure legend (lines 864-865 (in Fig. 1 legend), lines 874-876 (in Fig. 2 legend), lines 883-885 (in Fig. 3 legend), line 897 (in Fig. 4 legend), lines 904-911 (in Fig. 5 legend), lines 920-923 (in Fig. 6 legend), line 941-945 (in Fig. 8 legend), and line 946-947 (in Fig. 9 legend). The n-values added to correspond to the number of flies examined. In addition, we also added the number of replicates there.

Minor:

1.“depicted” RNA interference should be “experienced” RNA interference, I think?

Considering the reviewer’s comment, we revised the phrase as follows; “the adult muscle harboring RNA interference of Polg (PolgRNAi).” (line 92).

2.sentence from lines 398-400 about copy number should be moved to the previous section where copy number is being discussed

We moved the sentence to the lines 331−332, accordingly. 

3.on line 469, it says “autophagy occurred in the mitochondria” I think this gives a mistaken impression. Maybe should say “autophagy occurred in muscle”

We revised the mistake correctively, as requested (line 398). We appreciate the reviewer for a careful reading of our manuscript.

4.on line 660, “alleviated” should be “accelerated” I think

We replaced the word as requested (line 538).

Reviewer 2 Report

In the manuscript entitled “Downregulating mitochondrial DNA polymerase γ in the muscle stimulated autophagy, apoptosis, and muscle aging-related phenotypes in Drosophila adults” the authors have targeted DNA polymerase γ in order to understand the role of mitochondrial DNA damage in accelerating age-related phenotypes in Drosophila. Through a series of experiments, the authors have shown that muscle-specific silencing of DNA pol γ results in the accumulation of abnormal mitochondria with reduced membrane potential and mitochondrial activity. In continuation, the authors have demonstrated that both autophagy and mitophagy were enhanced in order to eliminate the damaged and abnormal mitochondria in the adult fly muscles. To support the findings, the status of apoptotic cells was also checked where an enhancement in cell death was observed in pol γ RNAi fly muscles. The authors have also shown that downregulation of the Pol γ genes in the muscle shortened the adult lifespan and accelerated the age-dependent locomotion impairment.

The results are interesting and the findings advance the field, particularly in the aging and mitophagy-related areas. 

However, there are some technical issues to be addressed to support the conclusions of the paper which are outlined below. 

·      In Figure 1, there is a discrepancy where the authors have mentioned the fly age as 36, and the graph shows day 30. The labeling in Figure 1 also needs to be revised. It’s confusing for the readers to understand as of now.

·      In Figure 2, the authors have shown the quantification of mtDNA in the indirect flight muscles of the PolG1RNAior PolG2RNAi adults fed paraquat. The results are interesting but the present magnification does not make the observation very convincing and it is difficult to discern whether there is a genuine enhancement in mtDNA levels.

·      To estimate mitochondrial activity, the authors investigated mitochondrial membrane potential using TMRE. n number should be provided for each representative picture.  

·      A high magnification picture for Figure: 4 and 5 could support the author’s hypothesis in a more convincing manner.

·      I think there is more background in images of 20-day-old PolGRNAi muscle samples. Are the images taken with the same microscope settings?

·      The authors have shown that downregulation of the Pol γ genes in the muscle shortened the adult lifespan. I am curious about the controls that the authors have used for this assay. Mef2ts>+ does not explain the controls used here. Since Drosophila wild-type flies show a significant variation in lifespans, for instance, CantonS have an average lifespan of 55-60 days whereas WDahomey has an average lifespan of 75-85 days. Therefore, proper control is important here.

·      Another important concern that I would like to raise is if the authors have tested whether PolG1RNAi over-expression can potentially increase the lifespan. It would be interesting to look if over-expressing PolG1RNAi in aged flies can improve mitochondrial status.

Author Response

In Figure 1, there is a discrepancy where the authors have mentioned the fly age as 36, and the graph shows day 30. The labeling in Figure 1 also needs to be revised. It’s confusing for the readers to understand as of now.

We corrected the mistake according to the reviewer’s request (line 857). We appreciate the reviewer for a careful reading of our manuscript.

In Figure 2, the authors have shown the quantification of mtDNA in the indirect flight muscles of the PolG1RNAior PolG2RNAi adults fed paraquat. The results are interesting but the present magnification does not make the observation very convincing and it is difficult to discern whether there is a genuine enhancement in mtDNA levels.

Considering the reviewer’s concern, we revised Figure 2 using original confocal micrographs of Picogreen-stained muscle samples to be able to visualize mitochondrial DNA in cytoplasm of muscle cells.

To estimate mitochondrial activity, the authors investigated mitochondrial membrane potential using TMRE.  n number should be provided for each representative picture.  

According to the reviewer’s request, we added the numbers of confocal images that quantified the TMRE fluorescence intensity in Figure 3 legend (lines 884-887). 

A high magnification picture for Figure: 4 and 5 could support the author’s hypothesis in a more convincing manner.

According to the reviewer’s request, we changed the layout of the images to magnify each panel in Figures 4 and 5 to show the details better.

I think there is more background in images of 20-day-old PolGRNAi muscle samples. Are the images taken with the same microscope settings?

We remade Figure 5A and 6A from original microscopic images. We acquired every image in Figure 5A or 6A using a confocal microscope with the same settings.

The authors have shown that downregulation of the Pol γ genes in the muscle shortened the adult lifespan. I am curious about the controls that the authors have used for this assay. Mef2ts>+ does not explain the controls used here. Since Drosophila wild-type flies show a significant variation in lifespans, for instance, CantonS have an average lifespan of 55-60 days whereas WDahomey has an average lifespan of 75-85 days. Therefore, proper control is important here.

We used the F1 progenies derived from a cross between a Mef2ts-Gal4 stock and a standard wild-type stock, Canton S, as control flies without expressing dsRNAs against PolG mRNAs. In addition, we briefly explained what the Mef2ts>+ means in Figure legend 1 (lines 859-860).

Another important concern that I would like to raise is if the authors have tested whether PolG1RNAi over-expression can potentially increase the lifespan. It would be interesting to look if over-expressing PolG1RNAi in aged flies can improve mitochondrial status.

We agreed that these two experiments recommended sound interest. We are more interested in the later experiment that examines whether overexpression of PolG1 in aged flies can improve the declined mitochondrial activity. We will address these two issues in our future study. We appreciate the reviewer’s kind suggestion. 

Round 2

Reviewer 2 Report

I would like to thank the authors for considering my comments. The authors have made significant improvements to the manuscript and answered all of my queries either through clarifications or by supplying additional data including quantification. The manuscript describes that muscle-specific silencing of DNA pol γ results in the accumulation of abnormal mitochondria with reduced membrane potential and mitochondrial activity. The findings support the conclusion.